# Dental Infection Requiring Hospitalisation Is a Public Health Problem in Australia: A Systematic Review Demonstrating an Urgent Need for Published Data

**DOI:** 10.3390/dj11040097

**Published:** 2023-04-04

**Authors:** Mafaz Ullah, Muhammad Irshad, Albert Yaacoub, Eric Carter, Andrew Thorpe, Hans Zoellner, Stephen Cox

**Affiliations:** 1Discipline of Oral Surgery, Faculty of Medicine and Health, University of Sydney, Sydney, NSW 2750, Australia; 2Nepean Centre for Oral Health, Nepean Hospital, Nepean Blue Mountains Local Health District, Kingswood, NSW 2747, Australia; 3Department of Oral Pathology, Rehman College of Dentistry, Peshawar 25000, Pakistan; 4Specialised Dental Center, Ministry of Health, Sakaka Aljouf 72345, Saudi Arabia; 5School of Nursing and Midwifery, Western Sydney University, Locked Bag 1797, Penrith, NSW 2751, Australia; 6Biomedical Engineering, Faculty of Engineering, The University of Sydney, Sydney, NSW 2006, Australia; 7Graduate School of Biomedical Engineering, University of NSW, Kensington, NSW 2052, Australia; 8Strongarch Pty Ltd., Pennant Hills, NSW 2120, Australia

**Keywords:** dental infection, dental caries, pericoronitis, hospitalisations, mortality, Australia

## Abstract

**Background**: The aim of this systematic review was to analyse the published literature on dental infections leading to hospitalisations in Australia. It was hoped that understanding the patterns and trends would form a basis for improved preventive and management policies. **Methods:** An electronic search was performed using Web of Science, Medline via Ovid and Google Scholar. Inclusion and exclusion criteria were applied. The included studies were analysed for demographics, aetiology, management, length of hospital stay and outcome of dental infections requiring hospitalisation. **Results:** Nine retrospective studies were eligible for inclusion. A total of 2196 cases of dental infections leading to hospitalisations were reported, with a male predominance (55–67%). Mental health issues, illicit substance abuse and immunosuppression were the main associated comorbidities (up to 58%). Dental caries (59–90%) and pericoronitis (10–19%) were the leading causes of dental infections. Empirical antibiotics were utilised in up to 75% of cases prior to hospital presentation. Six mortalities were reported. **Conclusions:** The available published data show that dental infection is a significant public health problem. However, only general conclusions were possible due to the variably small sample size and data collection that was inconsistent and incomplete across studies. Improved data collection is required to develop policies for prevention and management.

## 1. Introduction

Dental infections are bacterial in origin and extend from the dental pulp within teeth to the alveolus, jaws or face, and are often described as “odontogenic infections” to distinguish them from infections from origins other than teeth. Dental decay is the most common bacterial disease, and although decay is initiated by bacterial biofilm on the surface of the tooth, if unchecked, bacteria penetrate firstly into the underlying dentine, and then sequentially into the vascular dental pulp, the adjacent jaw bone and then neighbouring soft tissues where there is potential for systemic sepsis [1]. Spreading sepsis may also arise from periodontitis, where there is inflammatory destruction of the supporting tissues of the teeth; from pericoronitis, where there is inflammation of tissues surrounding partially erupted teeth; and from dental trauma [1,2,3,4,5]. Given this, it is unsurprising that up to 78% of all head and neck infections are of dental origin [6]. A wide range of medical conditions exacerbate dental infection, including diabetes mellitus, immunosuppression, chronic alcohol abuse, obesity, liver disease, chemotherapy and radiotherapy [2,7]. In addition, socioeconomic disadvantage is associated with increased dental caries and periodontitis, and hence with increased dental infection [8].

Dental infections may compromise airways and cause cavernous sinus thrombosis, orbital abscesses, cerebral abscesses, necrotizing fasciitis, mediastinitis or multi-organ failure and death [7]. In Australia, approximately 37% of all adult dental emergency visits to public hospital emergency departments are due to dental infections [9,10]. Coronial inquests from dental-infection-related mortalities have identified airway obstruction as an important cause of death with dental infection following surgery, pharyngeal abscess and Ludwig’s angina [11]. The death of the famous young Australian boxer Les Darcy whilst visiting the United States is a notable historical example of dental-infection-related mortality [12].

Dental infection is the leading dental-related cause of potentially preventable hospitalisations (PPH) in Australia [13]. Dental-related hospitalisations are primarily due to dental decay and account for 10% of total PPH and 22% of PPH due to acute conditions [13]. Notably, there was an increase in dental PPH from 57,955 in 2007–2008 to 72,000 in 2017–2018 [13,14]. Dental decay and impacted teeth were the most common causes of hospitalisations in Australia reported in a recent systematic review; however, most of the studies were from Western Australia [9]. Dental-related hospitalisations are a considerable and increasing financial burden on the health care system of Australia [15]. A recent study [16] reported 316,937 PPH for dental conditions over five years (2011–2015). Another study from Western Australia reported that over the 10-year period studied (1999–2009), more than 130,000 adults required hospitalisation for dental conditions at a direct cost of more than AUD 400 million [17]. A separate report on the financial burden of odontogenic infections showed that 462 cases of dental infections between 2006 and 2014 required hospitalisation, and the average cost per patient was AUD 12,228 [18].

Most dental decay and periodontal disease are preventable with regular dental service, patient education and timely management of early disease [19]. Similarly, timely and appropriate management can localise and resolve most dental infections in the early stages without complications. Management of dental infection requires removal of the cause, which may involve tooth extraction or root canal treatment. Incision with drainage and supportive therapy including antimicrobials, rehydration, nutrition and rest are also routine for the management of dental infection [20].

From the above, it is clear that dental infection leading to hospitalisation is a common and important public health problem in Australia. We aimed to conduct a systematic review of the available literature, with the expectation that it might be possible to achieve clarity on key aspects of dental infection across Australia that may be required for planning and development.

## 2. Materials and Methods

### 2.1. Review Question

The current review sought to answer the question, “What are the demographics, aetiology, management, hospitalisations and outcome of dental infections requiring hospitalisation in Australia, from the available published literature?” Preferred Reporting Items for Systematic Reviews and Meta-Analysis (PRISMA) guidelines were applied to establish this question, as opposed to the often-used Patient Intervention Comparison Outcome (PICO) search strategy tool, because our interest was not to explore any specific intervention as normally required by PICO, but rather to establish the dimensionality of relevant demographic and clinical information [21,22].

### 2.2. Search Strategy

The search terms were generated by using different combinations of the National Library of Medicine’s medical subject headings, all of which were PubMed MeSH terms. A bibliographical search without date restrictions was performed using Web of Science, Medline via Ovid and Google Scholar with the help of relevant keywords. The date of the last search was 31 May 2022. Additional articles were found following a search of the reference list of the articles found in the initial search as well as a search of Google Scholar using the same keywords.

The search line was “Periapical abscess/or Periodontal abscess/or Retropharyngeal abscess or Tooth abscess*.mp. or Dental abscess*.mp. or Odontogenic abscess*.mp. or Dentoalveolar abscess*.mp. or Jaw abscess*.mp. or Maxillofacial abscess*.mp. or Mandibular abscess*.mp. or Maxillary abscess*.mp. or Dental focal infection*.mp. or Dental infection*.mp. or Periapical infection*.mp. or Maxillofacial infection*.mp. or Mandibular infection*.mp. or maxillary infection*.mp. or Oral Infection*.mp. or Periodontal infection*.mp. or Odontogenic infection*.mp. or Pericoronitis AND exp Australia/or Australia*.mp.”.

### 2.3. Study Selection and Data Extraction

All records were collected into a single Endnote library and duplicates were removed. Titles and abstracts were screened by two independent reviewers (M.U. and M.I.). Specific publications were selected for full-text reading, according to the inclusion and exclusion criteria given below. Any disagreements between the reviewers were settled by discussion together with a third reviewer (S.C.). Relevant data were obtained and analysed by the first author (M.U.).

Inclusion criteria: Studies published in English language, addressing the review question, original hospital-based and Australian population-based studies and studies reporting both genders with a minimum study duration of 12 months.

Exclusion criteria: Studies conducted outside Australia, review studies, case report or case series studies, abstract only, letters, textbooks and interviews, conference papers and surveys.

### 2.4. Quality Evaluation

Quality appraisal of the selected articles was performed using the Mixed Method Appraisal Tool [9,23]. The studies were assessed against six criteria to evaluate the quality of each publication, and a final quality score was calculated for each publication. All studies fulfilled the MMAT criteria.

The PRISMA flow of the literature search and publication selection process is shown below [21].



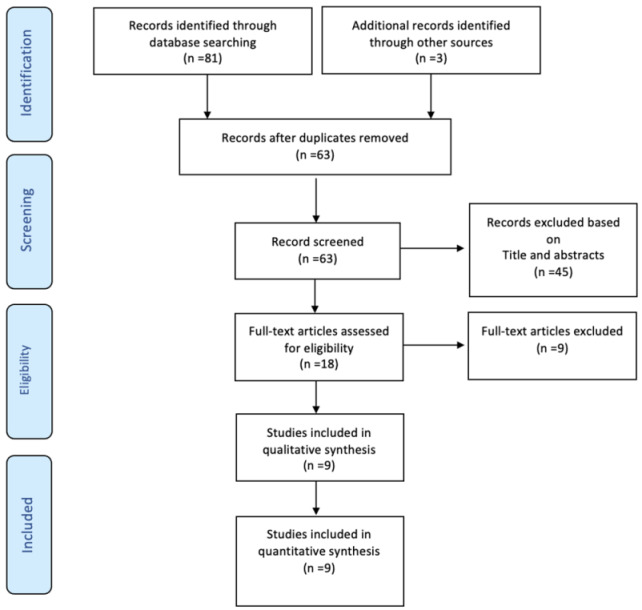



## 3. Results

### 3.1. Search Outcomes

After the removal of duplicates, a total of 63 papers were identified as being eligible for inclusion. A total of 18 publications were identified for full-text reading, resulting in the final selection of 9 studies based on the inclusion and exclusion criteria. Reasons for the exclusion of the other nine studies were as follows: interviews/surveys/review studies with no data, case reports/series studies, duration of studies less than 12 months. A quality appraisal of included studies using the Mixed Method Appraisal Tool is provided in the Appendix A.

### 3.2. Features of the Included Studies

The nine studies included were all retrospective and published between 1995 and 2020 (Table 1). Four studies were reported from South Australia (SA) [5,18,24,25], two from Queensland (QLD) [26,27], and one study each from Victoria (VIC) [3], New South Wales (NSW) [4] and Tasmania (Tas) [10]. A total of 2196 cases of dental infection were reported in the nine studies, and the number of cases ranged from 48 to 672 per study (Table 2). Two publications utilized the same data [26,27]. Dental infections were reported to be the main cause of maxillofacial infections (55–85%) [3,5,24].

### 3.3. Demographic Details

Six studies (Table 2) reported dental infections in adults with an age range of 16 to 88 years (mean 34 to 39) [5,10,18,24,25,26,27], although one study reported facial swelling in children under 14 years (mean 6.3) [4] and another study had a mean age of patients of 23 years [3]. Gender was considered in all studies, reporting 55.5% to 67% males. Aboriginal and Torres Strait Islander status was reported in only three studies and ranged from 5% to 17% [5,24,26,27].

### 3.4. Aetiology

All studies reported infection originating from dental tissues [3,4,5,10,18,24,25,26,27]; three specified if the origin was pulpal (ranging from 59% to 90%) or pericoronitis (ranging from 10% to 19%) [3,4,5] (Table 2).

### 3.5. Co-Morbidities for Odontogenic Infections

Four studies reported on comorbidities (up to 58%) (Table 2) [4,5,24,26,27], including mental health issues, substance abuse, diabetes mellitus and immunosuppression. In children, the associated comorbidities were mainly developmental delay and congenital heart conditions [4]. Only one study reported on the smoking status and found that 52% of patients were smokers [26,27].

### 3.6. Facial Space Involvement

Facial spaces were reported in only three studies [3,4,5]. Multiple spaces were involved in 44% to 47% of cases, although in children under the age of 14 years, 11% of cases were reported with multiple space involvement (Table 2). The most common spaces involved were buccal (26% to 59%) and submandibular (14% to 38%).

### 3.7. Microbiology

Microbiology was reported in three studies [3,5,25], recording mixed aerobic and anaerobic streptococci, methicillin-resistant staphylococcus aureus (MRSA), alpha haemolytic streptococci, bacteroides species and actinomyces israelii (Table 2).

### 3.8. Management of Dental Infections

Four studies detailed surgical management (Table 3), administered to 75% to 100% of cases [3,5,18,24], and involving tooth removal and drainage mostly under general anaesthesia, ranging from 63.5% to 87%. Incision and drainage were either extraoral (9% to 83.1%) or intraoral (16.9% to 59%), and sometimes both (32%). Five studies reported on antibiotic management [3,4,5,18,26,27]. The combination of penicillin and metronidazole with or without beta lactam (82.5% to 84%) was predominantly used, and the mode of administration was intravenous. For patients allergic to penicillin, cephazolin was often used [5]. Other types of antibiotics used included piperacillin and tazobactam for broadened coverage, or definitive antibacterial therapy [26,27]. The first choice of oral antibiotics at discharge was also penicillin [4,5].

### 3.9. Hospitalisations

Six studies indicated that all patients who presented with severe dental infections required hospitalisation (Table 3), while a separate study reported that 21.3% of patients required hospitalisation [10]. The study of dental infections in children reported that 14.6% of cases required hospitalisation [4]. The mean length of stay for adult patients ranged from 1 to 21 days (mean 2.6 to 4.18). ICU admissions were reported in 30% to 100% of cases.

### 3.10. Previous Treatment

Previous treatment, including antibiotics and extractions, was reported in four studies and ranged from 33% to 74.7% of cases, the highest being in children with facial swelling [3,4,5,26,27]. Previous treatment comprised short- and long-term courses of antibiotics, dental extractions with or without attempted drainage, dentist and general practitioner consultations and self-treatment.

### 3.11. Mortality

Six patients were reported from two studies as having died in Australia following dental infections between 2006 and 2014 (Table 3) [24,25]. Three deaths resulted from dental infections associated with sepsis with multiorgan failure, cavernous sinus thrombosis and necrotizing fasciitis, all involving multidrug-resistant organisms. The fourth, an elderly patient, died due to ventilator-associated pneumonia following management of dental infection. Details of terminal events resulting in death from dental infection were not provided for the remaining two patients [24].

## 4. Discussion

The current study identified some general aspects of dental infections with hospitalisation in Australia, and also identified important gaps and a need to establish improved data-gathering mechanisms.

More males presented with dental infections requiring hospitalisations, which is consistent with global data [28,29,30]. Similarly, the mean age at presentation for adult patients is in the thirties, which is in line with the data reported in the United Kingdom and the United States of America [28,29]. Only three studies reported Aboriginal and Torres Strait Islander status and the socioeconomic status of the patients. Unfortunately, the data did not include accessibility and remoteness, nor is there a report on demographic details such as urban as opposed to rural location, indigenous compared to non-indigenous origin, as well as socioeconomic status, which would be informative.

Four studies included in this review provided information concerning comorbid conditions [4,5,24,26,27], including mental health issues, drug abuse and immunocompromised status. It is well known that patients presenting with dental infections and associated comorbidities are at risk of serious infections and extended LOS in hospital and ICU [7,29,31,32,33]; thus, it is imperative to report all the coexisting conditions, including patients with disabilities and smoking. People with mental health problems have more decayed, missing and filled teeth compared with healthy people [34]. Furthermore, illicit drug users are more likely to present with severe dental caries and periodontal diseases and have reduced access to oral health care [35]. Similarly, patients who present with dental infections and uncontrolled diabetes mellitus and immunosuppression are at risk of developing sepsis [7,36]. It is reasonable to suggest that individuals in these groups require greater attention and preventative oral health measures.

Among the included studies, only three [3,4,5] specified the etiological origin of dental infections, most commonly dental caries, followed by pericoronitis. Interestingly, two studies from one cohort [26,27] found that third molar infection was the main cause of dental infections requiring ICU admission. This is consistent with the anatomical location and relation of the last two mandibular molar teeth [37]. The National Study of Adult Oral Health 2017–18 reported that one in every three Australian dentate adults had untreated dental caries [38]. Factors related to the prevalence of untreated dental caries in the Australian population include gender, level of schooling, indigenous status, socioeconomic position, rural or remote location, access to public oral health and dental insurance [38,39,40].

In this analysis, most cases of dental infections required removal of the cause of infection, such as extraction of the offending tooth and incision with drainage. The empirical use of penicillin with metronidazole antibiotics was predominant. A significant proportion (33–75%) of patients received prior treatment from a general medical practitioner or dentist, mainly in the form of antibiotic prescription. It is difficult to assess if the use of antibiotics was adjunctive or an independent treatment. The FDI World Dental Federation has highlighted the emerging issues of antibiotic resistance and advocated restricting its use to only when absolutely required [41]. This suggests an alarming situation where antibiotics are prescribed without definitive management or without removing the source of infection. Unable to access dedicated dental services, many with dental infection seek and receive antibiotics from medical practitioners supported by Medicare [3,4,5], but this fails to provide the definitive dental surgical management needed and feeds progressive community antibiotic resistance [25].

Increased morbidity and mortality are associated with dental infections caused by multidrug-resistant microorganisms [25]. For three out of six mortalities reported in this analysis, multidrug-resistant microorganisms were identified in cultures of dental infections [25]. One study reported the morbidity associated with dental infections, including ventilator-associated pneumonia 8% (17), septic shock 6% (12) and mediastinitis 1% (2) [24]. Three studies [3,5,25] reported microbiological data, and these indicate mixed aerobic with anaerobic infection representing the normal oral flora. One study reported the total rate of resistance to any antibiotic as 17.8%, while the rate of resistance to penicillin was 9.7% [25]. The presence of antibiotic resistance in patients with odontogenic infections was corelated with longer hospital and ICU LOS [25].

The limitations of the current study include the heterogenicity of published data, the retrospective design and the paucity of original research about dental infections across all states of Australia. Furthermore, the significant variations between the studies presented limit the ability to develop a clear picture of the severity of the situation in Australia. For these reasons, only general conclusions regarding the health burden of dental infections are possible.

The current study appears to be unique in that we have not found similar systematic reviews on preventable dental hospitalisations in Australia or for other nations. Nonetheless, individual studies of emergency presentations are reported, and it is interesting to observe that the number of people affected in the United Kingdom [28] is two to three orders of magnitude less compared to the United States [42,43]. This may reflect the impact of the United Kingdom’s National Health Service, which does support dental care, as opposed to circumstances in the United States that are more similar to those in Australia with regard to dentistry.

Medicare, Australia’s national health insurance scheme excludes the adult population from the regular dental care required to prevent and limit dental infection [44]. While a proportion of the population possessing government concession cards are eligible for state-run public dental services, prolonged waiting lists and times preclude timely access, with consequent dental infection requiring hospitalisation [45,46,47]. Patients suffering medical frailty and or clinically significant syndromes are likely most severely affected [48], although explicit data on this are lacking in the current study. The cost of dentistry is thus a significant barrier for accessing dental care in Australia. As a result, despite most dental infections being easily prevented or treated at an early stage, the cost barrier blocks access to preventive and early treatment for many people. The Australian Medicare system introduced a Chronic Dental Disease Scheme for chronically ill patients in 2007; however, it was closed in 2012 as a result of combined political events and probable mismanagement [49]. The Child Dental Benefits Schedule introduced in 2014 provides AUD 1000 over two years for the dental care of children aged between 2 and 17 years for eligible families [50]. However, only 22% of eligible children accessed the scheme in 2014, with a further decline of 16.3% the following year [50]. This likely reflects low clinical need amongst children in this well-fluoridated population.

## 5. Conclusions

In conclusion, some keys aspects of dental infections requiring hospitalisation have been identified; however, the variability in the details recorded for dental infection requiring hospitalisation makes it impossible to properly quantify the magnitude of the problem. Not only are there qualitative differences in the type of data captured, but the sample size is often too small for confident quantification, and this is reflected by the wide range of results for many parameters of interest. While this study demonstrates that hospitalisation due to dental infection is a significant public health problem, it is also clear that current data collection is insufficient to properly evaluate important aspects for policies on prevention and management to be properly developed. A systematic nationwide approach is required to collect suitably complete and consistent data on dental infections requiring hospitalisation in order to develop effective policies.

## Figures and Tables

**Table 1 dentistry-11-00097-t001:** Characteristics of the included studies.

Year	Author	State	Hospital	Study Type	Study Duration (Months)
1995	Bridgeman et al. [3]	Vic	The Royal Melbourne Hospital	Retrospective (1988–1992)	57
2005	Uluibau et al. [5]	SA	The Royal Adelaide Hospital	Retrospective (2003)	12
2014	Michael and Hibbert [4]	NSW	The Children’s Hospital at Westmead	Retrospective (2012–2013)	12
2014	Verma and Chambers [10]	Tas	The Royal Hobart Hospital	Retrospective (2012)	12
2015	Sundararajan et al. [24]	SA	The Royal Adelaide Hospital	Retrospective (2008–2013)	60
2018	Liau et al. [25]	SA	The Royal Adelaide Hospital	Retrospective (2006–2014)	108
2018 2020	Fu et al. [26] Fu et al. [27]	Qld	The Royal Brisbane and Women’s Hospital	Retrospective Group (1) 2002–2004 Group (2) 2013–2014 (Two publications from same data)	Group (1) 12 Group (2) 12
2020	Han et al. [18]	SA	The Royal Adelaide Hospital	Retrospective (2006–2014)	108

Vic, Victoria; SA, South Australia; NSW, New South Wales; Tas, Tasmania; Qld, Queensland.

**Table 2 dentistry-11-00097-t002:** Demographics, aetiology, features and microbiology of dental infections.

Publication	Co-Morbidities	Aetiology	Facial Space Involvement	Microbiology
Bridgeman et al., 1995 [3]	NA	**Dental** **76.4%** Pulpal 59% Pericoronal 10% Postoperative 24% Infected cyst 7% **Non-dental** **23.6%** Trauma 8.5% Fixation devices 7.5% Bone graft 7.6% Others	**Multiple spaces 47%** Submandibular 82% Buccal 56% Sublingual 36% Submasseteric 32% Submental 32% Others **Single space 53%** Buccal 53% Submandibular 25% Infraorbital 14% Others	**C&S 75.7% ** *No growth 3.7%* *Mixed oral flora 80.2%* MRSA *6.3%* a-haemolytic streptococci 3.7 Bacteroides ssp 3.7 *Actinomyces israelii 2.4*
Uluibau et al., 2005 [5]	**Total 58%** Mental illness 19% Substance abuse 15% DM 8% Immunosuppressed 2% Drug Allergy 10%	Pulpal 81% Pericoronal 19%	Multiple spaces 44% Single space 54%	**Mixed oral flora** MRSA 2%
Michael and Hibbert, 2014 [4]	**Total 7%** Developmental delay 4% Developmental delay and congenital cardiac disease: 1.5% Others: 1.5%	Pulpal 90% Trauma 9% Dental anomaly 1%	**Multiple spaces 11%** **Single space 89** Buccal 59% Submandibular 14% Upper lip 13% Canine fossa 11% Others 3%	NA
Verma and Chambers, 2014 [10]	NA	Dental origin	NA	NA
Sundararajan et al., 2015 [24]	Mental illness 12% DM 11.5% Substance abuse 8.5%	Dental origin	NA	NA
Liau et al., 2018 [25]	NA	Dental origin	NA	**Superficial infections (n = 107)** Aerobic 43.9% Anaerobic 64.5% Microaerophilic 23.4% **Deep infections (n = 340)** Aerobic 40.3 Anaerobic 70.3% Microaerophilic 46.8%
Fu et al., 2018 [26] Fu et al., 2020 [27]	DM 7% Immunosuppressed 2% Smoker 52%	Dental origin	NA	NA
Han et al., 2020 [18]	NA	Dental origin	NA	NA
**Publication**	**Dental Infections**	**Gender**	**Age (Years)**	**ATSI and non-ATSI Status**
Bridgeman et al., 1995 [3]	**Maxillofacial infections 107** * Dental 82 (74.4%) * * Non-dental 25 (23%) *	M 60 (56%) F 47 (44%)	Mean 23 Range 16–83 Age 21–40 (65%)	NA
Uluibau et al., 2005 [5]	**Maxillofacial infections 88** * Dental 48 (55%) * * Non-dental 40 (45%) *	M 32 (67%) F 16 (33%)	Mean 34.5 Range 19–88	Aboriginal 8 (17%) Caucasian 36 (75%) Other 4 (8%)
Michael and Hibbert, 2014 [4]	253	M 141 (56%) F 111 (44%)	Mean 6.3 Range 1–14	NA
Verma and Chambers, 2014 [10]	169	M 106 (62.7%) F 63 (37.3%)	Mean 36.59 Range 0–86	NA
Sundararajan et al., 2015 [24]	**Maxillofacial infections 256** * Dental 218 (85.2%) * * Non-dental 38 (14.8%) *	M 159 (62%) F 97 (38%)	Median 37 Interquartile range 27–47	ATSI 28 (13%)
Liau et al., 2018 [25]	672	NA	NA	NA
Fu et al., 2018 [26] Fu et al., 2020 [27]	Group (1) 101 Group (2) 191	**Group (1) ** M 59 (58%) F 42 (42%) **Group (2) ** M 108 (57%) F 83 (43%)	**Group (1) ** Mean 34 ± 13.8 **Group (2) ** Mean 37.2 ± 14.4	**Group (1)** ATSI 5 (5%) **Group (2)** ATSI 13 (7%)
Han et al., 2020 [18]	462	M 252 (55.5%) F 210 (45.5%)	Mean 39.5 ± 15.9	NA

DM, diabetes mellitus; NA, not applicable; C&S, culture and sensitivity; MRSA, methicillin-resistant staphylococcus aureus. M, male; F, female; ATSI, Aboriginal and Torres Strait Islander.

**Table 3 dentistry-11-00097-t003:** Management, hospitalisation, previous treatment and mortality associated with dental infections.

Author	Management		Hospitalisation	Previous Treatment	Mortality
	Surgical	ABX			
Bridgeman et al., 1995 [3]	**Non-surgical 15 (14%)** **Surgical 92 (86%)** *I&D* I/O 54 (59%) E/O 8 (9%) E/O and E/O 30 (32%) **Anaesthesia** GA 68 (63.5%) LA 24 (24.4%)	**Empirical ABX** β lactam plus mtz **Definitive ABX** C&S guided	**LOS (days)** Mean 5 Range 1–21	**Total 47 (44%)** (Dental and GMP) ABX only 30 (63.8%) Attempted drainage 9 (19.1%) Extraction of teeth 7 (14.8) Combination 1 (2.1%)	Nil
Uluibau et al., 2005 [5]	**I&D 97% (47)** E/O 81% I/O 18% I/O via socket 2% **Ext. 90%** **Anaesthesia** GA 87% LA/Sedation 13%	**I/V** PCN-G plus mtz 70% **PCN allergy ** Cephazolin plus mtz 17% **ABX on discharge** AMOX	**LOS (days)** Mean 3.33 Range 1–16 **Remained intubated 40% (19)** ICU 31% (15) HDU 8% (4)	**Total 16 (33%) ** Dental 8 (17%) ABX only 7(15%) Self-treatment 1(2%)	Nil
Michael and Hibbert, 2014 [4]	NA	**IV (n = 39)** PCN-G plus mtz 84% PCN-G 7% Cephazolin plus mtz 3% Others 6% **ABX on discharge (n = 194)** *Single therapy (n = 124) 64%* AMOX 88%. Augment 6%. Cephalexin 4% mtz 2% *Dual Therapy (n 70) 36%* AMOX plus mtz 94% Augmentin plus mtz 3% Others 3%	**LOS (days)** (Total admissions 37) Mean 2.6 Range 1–5	Antibiotics 189 (74.7%) Previous facial swelling 187	Nil
Verma and Chambers, 2014 [10]	NA	NA	Admissions 21.3% ESSU 2.4%	NA	Nil
Sundararajan et al., 2015 [24]	Ext. 164 (75%)	NA	**Hospital Admission 100%** Hospital LOS (days) Mean 3.6 IQR 2.7–5.6 **ICU 218 (100%)** *ICU LOS (days) * Mean 1.5 IQR 0.7–2.6 *Readmission to ICU 22/218* *Re-exploration 6% (13*) *Tracheostomy 2% (5)* *Ventilation 22% (48)*	NA	2
Liau et al., 2018 [25]	NA	NA	**Admissions 100% (672)** Hospital LOS (days) Mean 4.18 **LOS ICU ** Mean 39.12 (hours) Return to theatre 30 (4.5%) **AMR Infections** *LOS Hospital (days)* Mean 8.97 *LOS ICU (Hrs*) Mean 88.55	NA	4
Fu et al., 2018 [26] Fu et al., 2020 [27]	NA	AMOX plus mtz 82.5% **Broadened spectrum** Piperacillin with tazobactam **TG guided ABX ** Group (1) 90.8% Group (2) 96.7%	**Hospital LOS (days)** *Group (1)* Mean (SD) 3.33 (2) *Group (2) * Mean (SD) 4.13 (4) **ICU admission** Group (1) 6.9% Group (2) 23.8%	**Dental ** Group (1) 42% Group (2) 45% **ABX** Group (1) 57% Group (2) 63%	Nil
Han et al., 2020 [18]	**GA** *I&D, Ext. 344 (74.5%)* E/O I&D 286 (83.1%) I/O I&D 58 (16.9%) **LA** I&D, Ext. 69 (14.8%)	IV ABX 49 (10.6%)	**Hospital LOS (days) ** Mean 4.0 ± 4.3 **ICU LOS (hours) ** 38.5 ± 60.4	NA	Nil

I&D, incision and drainage; I/O, intraoral; E/O, extraoral; GA, general anaesthesia; LA, local anaesthesia; Ext., extraction; NA, not applicable; I/V, intravenous; ABX, antibiotics; mtz, metronidazole; C&S, culture and sensitivity; PCN, penicillin; AMOX, amoxicillin; TG, therapeutic guideline; ICU, intensive care unit; HDU, high dependency unit; ESSU, emergency short stay unit; LOS, length of stay; IQR, interquartile; OT, operator theatre; GP, general medical practitioner.

## Data Availability

Not applicable.

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
