# Peer review of "Dental Infection Requiring Hospitalisation Is a Public Health Problem in Australia: A Systematic Review Demonstrating an Urgent Need for Published Data"

_dentistry, 2023, doi:10.3390/dj11040097_

Round 1

Reviewer 1 Report

Literature review generally well done, with some criticisms noted below:

-check that all keywords are pubmed mESH terms

-in the introduction section some general considerations on vulnerabilities should be added, with particular reference to health ones; in this regard, I recommend inserting the following scientific work in the reference section:

Cianetti S, Valenti C, Orso M, Lomurno G, Nardone M, Lomurno AP, Pagano S, Lombardo G. Systematic Review of the Literature on Dental Caries and Periodontal Disease in Socio-Economically Disadvantaged Individuals. Int J Environ Res Public Health. 2021 Nov 24;18(23):12360. doi:10.3390/ijerph182312360. PMID: 34886085; PMC ID: PMC8656978.

- define the PICOs strategy clearly in the strategy search

-from table 1 remove the titles of the works; they are useless

- the discussion must start from the general problem and not from the limits of the study

Some considerations, for example on frail or syndromic patients, should be made in the light of the recent literature. In this regard, I recommend inserting the following scientific work in the references:

Carli E, Pasini M, Pardossi F, Capotosti I, Narzisi A, Lardani L. Oral Health Preventive Program in Patients with Autism Spectrum Disorder. Children (Basel). 2022 Apr 10;9(4):535. doi:10.3390/children9040535. PMID: 35455579; PMCID: PMC9031336.

Author Response

Reviewer 1

Concern 1:

 -check that all keywords are pubmed mESH terms

Response to Concern 1:

We agree with the Reviewer on the importance using appropriate keywords, and confirm that all of the terms used were Pubmed MeSH terms.

To clarify this in the manuscript, we have included the following phrase in the materials and methods 'all of which were Pubmed MeSH terms', as shown below.

Search Strategy:  'The search terms were generated by using different combinations of National Library of Medicine, Medical Subject Headings, all of which were Pubmed MeSH terms.'

Concern 2:

-in the introduction section some general considerations on vulnerabilities should be added, with particular reference to health ones; in this regard, I recommend inserting the following scientific work in the reference section: Cianetti S, Valenti C, Orso M, Lomurno G, Nardone M, Lomurno AP, Pagano S, Lombardo G. Systematic Review of the Literature on Dental Caries and Periodontal Disease in Socio Economically Disadvantaged Individuals. Int J Environ Res Public Health. 2021 Nov 24;18(23):12360. doi:10.3390/ijerph182312360. PMID: 34886085; PMC ID: PMC8656978.

Response to Concern 2:

We thank the Reviewer for making this helpful suggestion to improve our manuscript, and have inserted the below sentence accordingly.

Introduction, Paragraph 1, lines 171-173: 'In addition, socioeconomic disadvantage is associated with increased dental caries and periodontitis, and hence with increased dental infection [8].'

Concern 3:

- define the PICOs strategy clearly in the strategy search

Response to Concern 3:

We thank the Reviewer for raising this point, because our reasons for using the PRISMA as opposed to PICO approach were not articulated in the original manuscript. 

Our interest in the current study, was to determine the dimensionality of clinical presentations and management approaches used, rather than to explore any specific intervention.  We do believe that the PRISMA guidelines better accommodate the purpose of the current study than the PICO approach.

To clarify this in the manuscript, we have substantially modified the relevant passage in the manuscript to now read as follows.

Review Question Lines 270-273:  'Preferred Reporting Items for Systematic Reviews and Meta-Analysis (PRISMA) guidelines were applied to establish this question, as opposed to the otherwise often used Patient Intervention Comparison Outcome (PICO) search strategy tool, because our interest was not to explore any specific intervention as normally required by PICO, but to instead establish the dimensionality of relevant demographic and clinical information [21,22].

Concern 4:

-from table 1 remove the titles of the works; they are useless

Response to Concern 4:

We have removed the relevant column from Table 1 as suggested by the Reviewer.

Concern 5:

- the discussion must start from the general problem and not from the limits of the study

Response to Concern 5:

We agree with the Reviewer that the structure of the Discussion was previously flawed, and that the generality of the study was not clearly stated in the original manuscript.

To correct this, the following paragraph has been shifted from the opening passage of the Discussion, to now appear latter in the Discussion as follows:

Discussion, Lines 539-542 :  'The limitations of the current study include the heterogenicity of published data, the retrospective design, and the paucity of original research about dental infections across all states of Australia. Furthermore, the significant variations between the studies presented, limit the ability to develop a clear picture of the severity of the situation in Australia. For these reasons, only general conclusions regarding the health burden of dental infections are possible.'

In addition, we have inserted the following paragraph at the start of the Discussion.

Discussion, First Paragraph Lines 478-480:  'The current study identified some general aspects of dental infections with hospitalization in the Australia, and also identified important gaps and a need to establish improved data gathering mechanisms.'

Concern 6:

- Some considerations, for example on frail or syndromic patients, should be made in the light of the recent literature. In this regard, I recommend inserting the following scientific work in the references: Carli E, Pasini M, Pardossi F, Capotosti I, Narzisi A, Lardani L. Oral Health Preventive Program in Patients with Autism Spectrum Disorder. Children (Basel). 2022 Apr 10;9(4):535. doi:10.3390/children9040535. PMID: 35455579; PMCID: PMC9031336.

Response to Concern 6:

We thank the Referee for suggesting this important aspect that was absent in the original manuscript.  We have addressed this in the revised manuscript by inclusion of the following in the Discussion.

Discussion Lines 552-553: 'Patients suffering medical frailty and or clinically significant syndromes are likely most severely affected [48], although explicit data on this is lacking in the current study.'

Reviewer 2 Report

Dear authors,

Congratulation for your interesting work and excellent writing style. I have the following curious points:

In the section 3.10 Mortality, the authors mentioned that six patients were reported, however the detail was given only four patients. Is the data of the other 2 missed? 

Since severe infection from the dental origin is preventable just by appropriate oral hygiene care, do you have any idea why it still happened in the well-equipped and well-educated oral hygiene self-care, knowledgeable dental-care personnel, and high dental care technology era?

Best regards

Author Response

Reviewer 2

Concern 1:

In the section 3.10 Mortality, the authors mentioned that six patients were reported, however the detail was given only four patients. Is the data of the other 2 missed?

Response to Concern 1:

We thank the Reviewer for noticing our having failed to include clarifying information on the two patients involved.  The Reviewer is correct in assuming that the requisite data was missing for those two patients.

To clarify this in the manuscript, we now include the following

Results, Line 475-476: 'Details of terminal events resulting in death from dental infection were not provided for the remaining two patients [24].

Concern 2:

Since severe infection from the dental origin is preventable just by appropriate oral hygiene care, do you have any idea why it still happened in the well-equipped and well educated oral hygiene self-care, knowledgeable dentalcare personnel, and high dental care technology era?

Response to Concern 2:

The Reviewer raises an important point.  In Australia, most dental service is in context of private dental practice, while access to publicly funded dentistry is restricted on basis of differing criteria across State jurisdictions, but is in general related to thresholds for low income and also child populations.  Approximately 40% of the population is eligible for public dentistry, but the public system employs only in the order of 10% of the dental workforce, so that few public patients are able to access routine dental care, with much of the treatment that is delivered in the public system focused emergency treatment. 

From the above, the cost of dentistry is the main barrier to access to dental services in Australia, including access to preventive oral hygiene services delivered by both dentists and oral hygienists.

To address this in the manuscript, we now include the following in the Discussion.

Discussion, Lines 554-456: 'The cost of dentistry is thus a significant barrier for accessing dental care in Australia. As a result, despite most dental infection being easily prevented or treated at an early stage, the cost barrier blocks access to preventive and early treatment for many people.'

To further clarify this aspect of the manuscript, we have modified other sections of the relevant paragraph to now read as follows:

Discussion, Lines 549-561: 'Medicare, Australia's National health insurance scheme, excludes the adult population from the regular dental care required to prevent and limit dental infection [44]. While a proportion of population possessing government concession cards are eligible for state run public dental service, prolonged waiting lists and times preclude timely access, with consequent dental infection requiring hospitalisation [45-47]. Patients suffering medical frailty and or clinically significant syndromes are likely most severely affected [48], although explicit data on this is lacking in the current study. The cost of dentistry is thus a significant barrier for accessing dental care in Australia. As a result, despite most dental infection being easily prevented or treated at an early stage, the cost barrier blocks access to preventive and early treatment for many people. The Australian Medicare system introduced a Chronic Dental Disease Scheme for chronically ill patients in 2007, however, it was closed in 2012 as a result of combined political events and probable mismanagement [49]. The Child Dental Benefits Schedule introduced in 2014 provides $1000 AUD over two years for dental care of children aged between 2 and 17 years for eligible families [50]. However, only 22% of eligible children accessed the scheme in 2014, with a further decline of 16.3% in the following year [50]. This likely reflects low clinical need amongst children in this well fluoridated population.'

Reviewer 3 Report

I evaluated the article titled: “Dental infection requiring hospitalisation is a major public health problem in Australia for which there is a paucity of published data: a systematic review”

Aim: analyse the available published literature about dental infections leading to hospitalisations in Australia

Title: it is not well done. rewrite it.

Abstract: Some parts are poorly written. Adjust it.

Intro, Results, and Conclusion: well done

M&M: Search strategy is not matching with the M&M from Abstract

Part of the results are in the M&M

Results: Where is the quality assessment/Risk of bias (table/figure)?

DISCUSSION: it could be better organized.

Please, include articles from other countries that had the same goal of this study (to compare results)

Author Response

Reviewer 3

Concern 1:

Title: it is not well done. rewrite it.

Response to Concern 1:

We were surprised by this concern raised by the Reviewer, but on reflection do see that the title was a little incomplete and clumsy, and also that our conclusion in the Discussion did not properly align with the original title. We have changed the title and modestly adjusted the conclusion accordingly, and hope that this properly addresses the Reviewer's concern.

New Title: 'A systematic review demonstrates that hospitalisation from dental infections is an Australian public health problem that requires improved data collection'

Conclusion, Last and first paragraph respectively:  'In conclusion, some keys aspects of dental infections requiring hospitalisation have been identified; however, the variability in the details recorded for dental infection requiring hospitalisation is such as to make it impossible to properly quantitate the magnitude of the problem. Not only are there qualitative differences in the type of data captured, but sample size is often too small for confident quantitation, and this is reflected by the wide range of results for many parameters of interest. While this study demonstrates that hospitalisation due to dental infection is a significant public health problem, it is also clear that current data collection is insufficient for policies on prevention and management to be properly developed. A systematic Nation-wide approach is required to collect suitably complete and consistent data on dental infections requiring hospitalisation, in order to develop effective policies.'

Concern 2:

Abstract: Some parts are poorly written. Adjust it.

Response to Concern 2:

We acknowledge the Reviewer's point, and have improved the abstract to more completely describe the scope, intent and conclusions of our study. 

The new abstract reads as follows:

'Background: This systematic review aimed to analyse the published literature on dental infections leading to hospitalisations in Australia. It was hoped  this would form basis for improved policies for prevention and management.

Methods: An electronic search was performed using Web of Science, Medline via Ovid and Google Scholar. Strict inclusion and exclusion criteria were applied The included studies were analysed for  demographics, aetiology, management, hospitalisation and outcome of dental infections requiring hospitalisation.

Results: Nine retrospective studies were eligible for inclusion. A total of 2196 cases of dental infections leading to hospitalisations were reported with a male predominance (55-67%). Mental health issues, drug abuse and immunosuppression were the main associated comorbidities (up to 58%). Dental caries (59-90%) and pericoronitis (10-19%) were the leading causes. The use of empirical antibiotics was up to 75% prior to presentation. Six mortalities were reported.

Conclusions: The available published data make it self-evident that dental infection is a significant public health problem. However, only general conclusions were possible due to variably small sample size, and data collection that was inconsistent and incomplete across studies. Improved data collection is required to develop policies for prevention and management.'

Concern 3:

M&M: Search strategy is not matching with the M&M from Abstract

Response to Concern 3:

We thank the Reviewer for detecting this lack of accuracy in the Abstract.

The materials and methods section of the abstract is now more complete and properly summarises the more detailed materials and methods provided in the body of the manuscript.

The new abstract section now reads as follows:

'Methods: An electronic search was performed using Web of Science, Medline via Ovid and Google Scholar. Strict inclusion and exclusion criteria were applied The included studies were analysed for  demographics, aetiology, management, hospitalisation and outcome of dental infections requiring hospitalisation.'

Concern 4:

Part of the results are in the M&M

Response to Concern 4:

We are glad that the Reviewer noticed our error, and have now significantly modified the relevant paragraph of the Materials and Methods, and also included a new paragraph in the Results, as shown below. 

Materials and Methods Lines 287-290: 'All records were collected into a single Endnote library and duplicates were removed.  . Titles and abstracts were screened by two independent reviewers (MU and MI). Specific publications were selected for full text reading, according to the inclusion and exclusion given below.. Any disagreements between the reviewers were settled by discussion together with  a third reviewer (S.C). Relevant data were obtained and analysed by the main author (MU).'

Results, Lines 359-363: 'After removal of duplicates, a total of 63 papers were identified as being eligible for inclusion. A total of 18 publications were identified for full text reading, resulting in final selection of 9 studies based on the inclusion and exclusion criteria. Reasons for exclusion of 9 studies were as follows: Interviews/surveys/review studies with no data, case reports/series studies, duration of studies less than 3 months.'

Concern 5:

Results: Where is the quality assessment/Risk of bias (table/figure)?

Response to Concern 5:

We thank the Reviewer for noting this omission.

We now provide the relevant table as supplemental information. 

Changes in the text reflecting this are as follows.

Results, Line 362-363: 'A quality appraisal of included studies using the mixed method appraisal tool is provided in Supplemental Materials (Table S1).'

Table provided in Supplemental Materials (Line 759)

Table S1. Quality appraisal of included studies using Mixed Method Appraisal Tool

Publication

Are there clear quantitative research questions (objectives)?

Do the collected data address the research question (objective)?

Are participants recruited in a way that minimises selection bias?

Are measurements appropriate regarding the exposure/ intervention and outcomes?

In the groups being compared, are the participants comparable, or do researchers take into account the difference between the groups?

Are there complete outcome data (80% or above), or an acceptable follow-up rate for cohort studies (depending on the duration of follow up)?

Overall quality score

Bridgeman

et.al., 1995[3]

Yes

Yes

Yes

Yes

Yes

Yes

1

Uluibau et.al., 2005[5]

Yes

Yes

Yes

Yes

Yes

Yes

1

Michael and Hibbert, 2014[4]

Yes

Yes

Yes

Yes

Yes

Yes

1

Verma and Chambers, 2014[10]

Yes

Yes

Yes

Yes

Yes

Yes

1

Sundararajan et. al., 2015[24]

Yes

Yes

Yes

Yes

Yes

Yes

1

Liau et. al., 2018[25]

Yes

Yes

Yes

Yes

Yes

Yes

1

Fu et al., 2018[26]

Yes

Yes

Yes

Yes

Yes

Yes

1

Fu et al., 2020[27]

Yes

Yes

Yes

Yes

Yes

Yes

1

Han et al., 2020[18]

Yes

Yes

Yes

Yes

Yes

Yes

1

Concern 6:

DISCUSSION: it could be better organized.

Response to Concern 6:

We agree with the Reviewer that improvement in the Discussion was required, and have made a number of changes that we believe address this concern.  Changes made are described below.

Insertion of a new first paragraph:

Discussion, Paragraph 1: 'The current study identified some general aspects of dental infections with hospitalization in the Australia, and also identified important gaps and a need to establish improved data gathering mechanisms.'

Shifting of the previous first paragraph to now appear later in the Discussion where it is more appropriate:

'The limitations of the current study include the heterogenicity of published data, the retrospective design, and the paucity of original research about dental infections across all states of Australia. Furthermore, the significant variations between the studies presented, limit the ability to develop a clear picture of the severity of the situation in Australia. For these reasons, only general conclusions regarding the health burden of dental infections are possible.'

Appreciable change to the last paragraphs of the Discussion so that it now read as follows.

Medicare, Australia's National health insurance scheme, excludes the adult population from the regular dental care required to prevent and limit dental infection [42]. While a proportion of population possessing government concession cards are eligible for state run public dental service, prolonged waiting lists and times preclude timely access, with consequent dental infection requiring hospitalisation [43-45]. Patients suffering medical frailty and or clinically significant syndromes are likely most severely affected [46], although explicit data on this is lacking in the current study. The cost of dentistry is thus a significant barrier for accessing dental care in Australia. As a result, despite most dental infection being easily prevented or treated at an early stage, the cost barrier blocks access to preventive and early treatment for many people. The Australian Medicare system introduced a Chronic Dental Disease Scheme for chronically ill patients in 2007, however, it was closed in 2012 as a result of combined political events and probable mismanagement [47]. The Child Dental Benefits Schedule introduced in 2014 provides $1000 AUD over two years for the dental care of children aged between 2 and 17 years for eligible families [48]. However, only 22% of eligible children accessed the scheme in 2014, with a further decline of 16.3% in the following year [48]. This likely reflects low clinical need amongst children in this well fluoridated population.'

The Conclusion has also been appreciably modified to more clearly express this aspect of discussion, and now reads as follows.

Conclusions: 'In conclusion, some keys aspects of dental infections requiring hospitalisation have been identified; however, the variability in the details recorded for dental infection requiring hospitalisation is such as to make it impossible to properly quantitate the magnitude of the problem. Not only are there qualitative differences in the type of data captured, but sample size is often too small for confident quantitation, and this is reflected by the wide range of results for many parameters of interest. While this study demonstrates that hospitalisation due to dental infection is a significant public health problem, it is also clear that current data collection is insufficient to properly evaluate important aspects for policies on prevention and management to be properly developed. A systematic Nation-wide approach is required to collect suitably complete and consistent data on dental infections requiring hospitalisation, in order to develop effective policies.'

Concern 7:

[In the Discussion] Please, include articles from other countries that had the same goal of this study (to compare results)

Response to Concern 7:

We are grateful that the Reviewer has suggested we discuss this aspect.  We find no equivalent systematic review for other countries.  However, there are individual studies that do address the overall issue, and we now include the below paragraph to address this in the manuscript. 

Discussion, lines 543-548:  'The current study appears to be unique in that we have not found similar systematic reviews on preventable dental hospitalizations in Australia or for other nations. Nonetheless, individual studies of emergency presentations are reported, and it is interesting to observe that the number of people affected in the United Kingdom (Carter and Layton) is two to three orders of magnitude less compared with in the United States (Shah et al;  Nalliah et al). This may reflect the impact of the United Kingdom's National Health Service which does support dental care, as opposed to circumstances in the United States that are more similar to those in Australia with regard to dentistry.'  

Round 2

Reviewer 3 Report

I evaluated the article titled: “A systematic review demonstrates that hospitalisation from dental infections is an Australian public health problem that requires improved data collection”

Aim: analyse the available published literature about dental infections leading to hospitalisations in Australia

Title: it is not well done. Please revise it again.

Nine articles included for a topic like that, I can consider insignificant. Please, I suggest to include continent or to use another criterium.

Why does the authors choose only hospitalization in Australia, whether you are performing a Systematic review??

Abstract: Some parts are poorly written yet.

Author Response

Covering note for the Editor

We thank the reviewers for their thoughtful and constructive advice.

We have addressed all of the concerns raised in the manuscript as shown in the detailed response to reviewers.

We hope that there will be agreement, that the manuscript is now suitable for publication.

With thanks again and very best regards,

Mafaz Ullah

Detailed Response Reviewer 3 report round 2

Concern 1: Title: it is not well done. Please revise it again.

Response: Thank you for the suggestion. It is reasonable to highlight the dental infection as a public health problem and there is limited published data. In view of this, we agree with the reviewer to improve the title to reflect the conclusion. We have changed the title which now reads as below.

Dental Infections Requiring Hospitalisation is a Public Health Problem in Australia: A Systemic Review Demonstrating An Urgent Need for Published Data

Concern 2: Nine articles included for a topic like that, I can consider insignificant. Please, I suggest to include continent or to use another criterium.

Response: We agree with the reviewer that limited published data is available on this topic in Australia which we have highlighted in the conclusion of this study as below.

In conclusion, some keys aspects of dental infections requiring hospitalisation have been identified; however, the variability in the details recorded for dental infection requiring hospitalisation is such as to make it impossible to properly quantitate the magnitude of the problem. Not only are there qualitative differences in the type of data captured, but sample size is often too small for confident quantitation, and this is reflected by the wide range of results for many parameters of interest. While this study demonstrates that hospitalisation due to dental infection is a significant public health problem, it is also clear that current data collection is insufficient to properly evaluate important aspects for policies on prevention and management to be properly developed. A systematic Nation-wide approach is required to collect suitably complete and consistent data on dental infections requiring hospitalisation, in order to develop effective policies.

The suggestion to include other countries is valid however the current inclusion criteria include only Australian studies which is unfortunately limited.

Concern 4: Why does the authors choose only hospitalization in Australia, whether you are performing a Systematic review??

Response: Oral health related conditions make a significant proportion of preventable hospitalisations. Among the oral health conditions, the highest proportion of preventable hospitalisation is due to dental infection. Preventable hospitalisation is a significant health burden on Australian economy. In the view of this we conducted a systematic review to analyse the Australian data so that we may find some trends and patterns which may be helpful to reduce preventable hospitalisation due to dental infection in Australia. This has been highlighted in introduction and discussion.

Concern Abstract: Some parts are poorly written yet.

Response: Thanks for highlighting some issues in the abstract which we have improved as below. 

Abstract:

Background: The aim of this systematic review is to analyse the published literature on dental infections leading to hospitalisations in Australia. It was hoped that understanding the patterns and trends would form basis for improved preventive and management policies.

Methods: An electronic search was performed using Web of Science, Medline via Ovid and Google Scholar. Inclusion and exclusion criteria were applied. The included studies were analysed for demographics, aetiology, management, length of hospital stay and outcome of dental infections requiring hospitalisation.

Results: Nine retrospective studies were eligible for inclusion. A total of 2196 cases of dental infections leading to hospitalisations were reported with a male predominance (55-67%). Mental health issues, illicit substance abuse and immunosuppression were the main associated comorbidities (up to 58%). Dental caries (59-90%) and pericoronitis (10-19%) were the leading causes of dental infections. Empirical antibiotics has been utilised in up to 75% of cases prior to hospital presentation. Six mortalities were reported.

Conclusions: The available published data make it self-evident that dental infection is a significant public health problem. However, only general conclusions were possible due to variably small sample size, and data collection that was inconsistent and incomplete across studies. Improved data collection is required to develop policies for prevention and management.

Round 3

Reviewer 3 Report

Dear authors,

thank you for all responses and adjusts.

Even though I considered limitations, your polite and correct responses were enough for me.

Congratulations!